

# Dendrochronologically dated pine stumps document phase wise bog expansion at a northwest German site between c. 6700 BC and c. 3400 BC

Inke Elisabeth Maike Achterberg[1], Jan Eckstein, Bernhard Birkholz, Andreas Bauerochse[2], Hanns Hubert Leuschner[1]

[1]Department for Palynology and Climate Dynamics, University of Göttingen, 37073 Göttingen, Germany

[2]Lower Saxony State Service of Cultural Heritage, 30175 Hannover, Germany

*Correspondence to*: Inke E. M. Achterberg (iachter@gwdg.de)

**Abstract.** This is a dendrochronological investigation of a mire site densely covered by peat-preserved pine stumps (*Pinus sylvestris*). The site in the northwest German *Tote Moor* revealed to feature trees from various Holocene millennia. The dendrochronologically dated site chronology covers 2345 years between 6703 BC and 3403 BC, containing 5 gaps between 6 and 550 years in length. It consists of 477 trees. A floating chronology segment of 309 years, containing 30 trees, was radiocarbon dated to the beginning of the 7th millennium cal. BC. The tree ring data from the site documents environmental changes over a larger period of time. Furthermore, the site is covered densely with in situ tree stumps from the fen-bog transition. This facilitates the spatio-temporal reconstruction of mire development, which is based on 212 in situ tree stumps in the case study presented here. Peat-stratigraphical survey was carried out additionally, and elevations a.s.l. were determined at several locations.

Tree die-off phases, which indicate the local water level rise, mostly in context of the local fen-bog transition, are evident for c. 6600-6450 BC, c. 6350-5750 BC, c. 5300-4900 BC, c. 4700-4550 BC, c. 3900-3850 BC, 3700-3600 BC, c. 3500-3450 BC and c. 3400 BC.

## 1 Introduction

Raised bog development shaped the northwest German lowland during the Holocene, as eventually about a third of the area had been covered by mires (fens and bogs) (Metzler, 2004). The development of these mires on the underlying nutrient poor glacial deposits was largely determined by climatic variations (Ellenberg, 1996). The area was particularly characterized by large lowland raised bogs, which grow better under humid and cool conditions (Behre, 2008). The expansion of these raised bogs is evident since the 7th millennium BC, with a strong increase between 5100 and 3600 BC (Eckstein et al., 2011, Petzelberger et al., 1999). The peat in the area contains a valuable environmental record of the past millennia.

The raised bogs were treeless in their central parts, while most of the surrounding region was wooded. Even the fens were wooded in large parts, often forming alder (*Alnus*) swamp forests or carrying other tree species. Often at the margins of a




raised bog a swamp (lagg) develops, fed by run-off water from the bog mixing with ground water. The more drained margin of the raised bog towards the lagg, is commonly wooded by pine (*Pinus*), in contrast to the otherwise generally treeless raised bog (Ellenberg, 1996). The expansion of the raised bogs often killed trees in the process and preserved their remains under *Sphagnum* peat. Stratigraphically the pine stumps mostly mark the local fen-bog transition. When in situ stumps are

dated, they document the raised bog expansion.

The large scale uncoverage of such stump layers through peat mining made areal dendrochronological investigation possible. The trees dated absolutely by dendrochronology deliver the finest temporal reconstruction of bog expansion and local bog formation respectively, particularly by their die-off dates (Eckstein et al., 2009, 2010, Edvardsson et al., 2014, Leuschner et al., 2002, 2007). Eckstein et al. (2009, 2010, 2011) have shown, that the tree die-off occurs in phases, which are often

synchronous in different mires. They were able to show that the tree die-off is mostly due to hydrological changes, while fire and storm have been found to be of little relevance on the investigated sites. Particularly the observed synchrony of pine establishment and die-off phases in different mires shows, that the local mire development documented by the trees is strongly linked to climatic variations (Eckstein et al., 2009, 2010, 2011, Edvardsson et al., 2011, 2012).

The phase-wise advance of a raised bog in northwest Germany is particularly well documented in the mire *Totes Moor* at the

site *TOMO-south* (Fig. 1). At the site, tree remains from one stratigraphical stump-layer (Fig. 2) turned out to originate from four millennia, documenting the bog progress across some 500 m.

At site *TOMO_south*, the spatial as well as the temporal distribution of many preserved trees has been obtained. The present study aims to connect areal changes to the temporal patterns observed.

## 2 Material and Methods

The *Tote Moor* is a mire located north of lake *Steinhuder Meer* near Hanover. The sandy slope below the peat appears to feature a net of rills and shallows, which probably held many little initiative cells of mire, that grew together over time. The whole depression of the lake is based on the same nutrient-poor sands that characterize the *geest* region.

On the site *TOMO_south* many in situ tree stumps, and also in situ stems and ex situ stumps (pulled at ditch-digging and left

at ditch-side) were sampled as radial slices using a chain saw. The tree remains were documented by using a feature table (regarding their growth, size, conservation condition etc.), photographic pictures and GPS coordinates. On some tree stumps, the root depth and shape was investigated. Later, the samples were dried, reduced by circular saw, and frozen. Then, suitable measuring radii were surface-cut by scalpel, contrasted by rubbed-in chalk dust and measured on an *Aniol* motorized measuring device with CATRAS measuring software to a precision of 0,01 mm. The tree ring-widths series were cross dated

using mainly the V-program-set (a.o. SYNCH2) by Thomas Riemer (Riemer, 1994 and unpublished work) and the *tsapWin* program by Rinntech (Rinn, 2003). The Lower Saxony chronologies of oak and pine were used as a base for dating. Part of the Lower Saxony Pine Chronology is also a product of this work. For the floating chronology segment three radiocarbon



dates, which were determined at the Leibniz Institute for Applied Geophysics (LIAG) in Hanover, were wiggle matched on base of the IntCal13 calibration curve (Reimer et al., 2013) using the *OxCal 4.2* online software (Bronk Ramsey et al., 2001, Bronk Ramsey, 2009).

*In situ* finds were later separated from ex situ finds. For the spatio-temporal reconstruction of raised bog development (Fig. 5), stumps with rootplates partially dragged upward, which retained part of their root system within the grown peat and thus their location, were used along with in situ s.s. finds. 96 trees have been possibly moved in such a way. They were added to the 116 trees which were considered 'in situ' in a stricter sense. In total 210 tree stumps with dendrochronological datings and 2 radiocarbon dated tree stumps are included in the reconstruction and displayed (Fig. 5 and 6). All 212 are referred to as 'in situ' in the following. All maps were created using ArcGIS 9 and 10 (Esri) mapping software. Figures were prepared using CoralDRAW software.

In addition, the peat-stratigraphy of 56 cores at the site was investigated macroscopically. Elevations a.s.l. were determined by stadia survey for 36 of the peat cores taken for this study, and for 63 on behalf of ASB-Humus peat mining company, who kindly shared their data. The elevation a.s.l. of the mineral base (sand) beneath the peat is depict as a regularized spline interpolation on base of the total 99 elevation measurements (Fig. 6).

Dendrochronological dates are given in years BC, radiocarbon dates in years cal. BC. Labelled time spans, like chronology segments or gaps, always include the years named.

## 3 Results

At the site *TOMO_south*, a rather levelled field left by peat mining revealed the remains of an apparent pine forest (Fig. 2). A closer look revealed, that what had appeared to be one continuous tree layer actually did contain neighbouring stumps grown on slightly different elevation levels (Fig. 3).

Of the 700 tree stumps sampled at this site most were pine (*Pinus sylvestris*), only 10 oak (*Quercus* spec.) stumps were sighted and sampled. The pine remains often had retained bark in the lower parts, often being preserved to bark edge or showing only minimal decay at the outermost rings.

### 3.1 Temporal distribution of the trees

Only after many pine stumps had been sampled, dendrochronological dating was successful for 477 trees. An additional floating chronology segment of 30 trees was radiocarbon dated. The trees at the seemingly homogeneous site in fact originated from various centuries (Fig. 4, Table 1). They grew around 7000 to 6700 +/-80 cal. BC (floating chronology segment) and between 6703 and 3403 BC (dendrochronological calendar dates). The floating chronology segment covers 309 years, the chronology segments with dendrochronological calendar dates cover 2345 years.

The majority of the trees (443) originates from the 7th to 5th millennia BC. Much less trees (34) represent the 4th millennium BC. Tree occurrence was not scattered over time, but is found to cluster in at least eight groups (chronology segments C14



and A1-F, Fig. 4). There are also periods without pine trees preserved at the site: while the two shortest of the five gaps in the site chronology (6 and 11 years in length) belong to the poorly replicated 4$^{th}$ millennium BC and are likely due to sampling design (which, in this case, is the placing of the site borderline through peat mining), the longer gaps in the well replicated chronology parts likely reflect the actual dynamic at the site.

There are periods where tree establishment and tree **die-off events** appear rather scattered (e.g. 6000-5800 BC or 5230-5120 BC), and intervals where the wooded phases display rather clustered tree establishment and tree die-off. Particularly clear are the die-off events around 6315 BC, 5060 BC and 3838 BC (Fig. 4).

Rejuvenation pulses, where several trees germinate within few years, are found within several segments. This can be seen
clearly in segments C14, A2 (repeatedly) and B. Less clearly segments C and E depict the same, while segment F lacks replication and no such pattern is found in segments A1 and D (which is also poorly replicated). A closer look reveals, that all rejuvenation pulses within segments are preceded by a time of limited or even missing rejuvenation. The rejuvenation pulses hence seem to mark the end of periods with limited tree establishment. In several cases the phases of limited tree establishment are also phases of frequent tree die-off . This is particularly clear for the germination pulse beginning with the
die-off event around 6315 BC.

### 3.2 Reconstructed bog expansion

Mapping of the dated in situ trees generally shows groups of contemporaneous trees growing together. The oldest trees, from the early 7$^{th}$ millennium BC, cluster in the south-east of the plot, while the youngest, those of the 4$^{th}$ millennium BC, are found to the north-west, with the rest arranged in between. This clearly shows the general direction of raised bog expansion
from **south-west to north-east** across the plot, even though there are some discontinuities within the site. Firstly, the groups of trees are not arranged in orderly stripes (Fig. 5), but form tongues and islands, often in correspondence to the dynamics of the mineral base (sand) beneath (Fig. 6). On sandy elevations trees persisted much longer, while the surrounding mire had long become a treeless raised bog (Fig. 7). Secondly, there are sometimes individual trees from one time interspersed into a group from another.

The mineral base is lowest to the south-east, where the oldest trees were found. Those trees document the oldest part of raised-bog in the plot.

In Fig. 5 the raised bog advance has been reconstructed according to the dated in situ tree remains. The interpolated areas (Fig. 5) were set in a 'no tree growth after' approach. That means: In places, where trees from different chronology segments occur intertwined, the place was assigned to the younger specimen.

This was done in an attempt to depict the advance of the raised bog, which, in its final state, is mostly treeless (Ellenberg, 1996).



### 3.3 Peat-stratigraphy

The peat stratigraphy is shown for 31 cores of the site (Fig. 8), displayed in elevational relation. In total, elevation a.s.l. has been determined for 36 of the 56 cores which were peat-stratigraphically investigated at the site. Please note, that some cores have been taken at the edge of the peat excavation field, where peat was left standing more than 1.5 m higher than the level
of the tree stump layer.

The general order above the glacial sand deposits is the following: brown moss peat on bottom, wood rich peat above, followed by *Sphagnum* peat on top. The pine tree remains are found at the fen-bog-transition. This picture is rather continuous over the site, with differences mostly restricted to layer thickness. The brown moss peat is mostly weakly humified, often features *Menyanthes* seeds (*Menyanthes trifolia*) in the middle or upper part, and silt near the bottom.
The wood rich peat varies, mostly being humified stronger, but containing in different spots and layers various portions of *Betula* (bark), *Alnus* (wood), *Pinus* (bark and cone) and charcoal.

A few spots feature an *Eriopherum* layer (*Eriopherum vaginatum*) below the *Sphagnum* peat. More cores show some *Eriopherum* mixed in with the lower *Sphagnum* peat or the upper wood rich peat.

The *Sphagnum* peat, where distinguishable, consists of *Sphagnum* section *Acutifolia* peat, *Sphagnum* section *Cuspidata* peat
and *Sphagnum-Carex* peat.

A few of the cores (two of the ones shown) feature highly humified fen peat in the lower part, which is likely to be brown moss peat in a decomposed state. In agreement with this assumption, the brown moss peat of most cores is more humified at its base.

### 3.4 Root morphology

There is one main feature, dividing the pines into two groups by their roots: the one type of root system spreads out flat, with no downward growing roots. The central root either died off at a length of 10 cm or less, or is not traceable at all. The other type of root system displays downward growing roots, usually with a strong central root growing straight down below the stem. Most often, these reach 20 to 40cm below the root plate.

These two types clearly show up in the ex situ material pulled from the ditches. Roots below the stem were investigated also
for 18 in situ stumps (Fig. 9), primarily to clarify if their roots reached the sand below the peat. This was not the case with the examined specimen.

### 4 Discussion

### 4.1 The preserved trees

During dry periods, with consequently lowered bog water tables, pine forestation can temporally also occur on the bog .
Such growth of large pines within raised bog (rooting in *Sphagnum* peat) was described i.a. by Overbeck (1954) for a mire in





northern Germany, by Moir et al. (2010) for such findings in Scotland, by Edvardsson et al. (2014) in southern Sweden, or in general terms by Ellenberg (1996).

However, this is not the case at *TOMO_south*, where the pines occur at the fen-bog transition and not within *Sphagnum* peat layers. As confirmed by peat stratigraphy, the tree remains in *TOMO_south* represent a persistent pine forestation at raised bog margin. Pine forestation at raised bog margins is a very typical occurrence. They colonize marginal parts of the bog, where the ascending bog surface is well drained compared to more central parts (Ellenberg, 1996, Overbeck, 1975).

The death and conservation of the trees however, appears to be closely connected to the expanding of the raised bog and the rise of the corresponding water table. This has been found evident on base of abundant upward growing roots late in the trees life, the drasticly narrowing rings near bark, peatstratigraphical context and the state of conservation at various comparable sites investigated in northwest Germany and southern Sweden (Eckstein et al., 2009, 2010, Edvardsson et al., 2012, 2014, Leuschner et al., 2002, 2007). Therefore the death of the trees and their conservation under *Sphagnum* peat neatly dates raised bog expansion, and using in situ stumps adds location to the event. Seeing the continuously dense (spatial) cover of the site with tree stumps (Fig. 2), all significant and lasting raised bog expansion should be documented by embedded trees. The tree data displays phases of cumulative die-off on the one hand and site chronology gaps on the other. This documents phases of the bog expanding, and phases without significant expansion.

The phases of bog expansion should generally relate to relatively humid climate phases, due their dependency on a high portion of rain water in the water present (Ellenberg, 1996). Raised bog formation can be favoured by a preceding lowering of water table through the disconnection of the fen peat surface from the lowered ground water and its nutrient supply (Hughes 2004, Tahvanainen, 2011, von Bülow, 1935). At site *TOMO_south* this combination of a drier phase with lowered water tables followed by a phase of swift raised bog expansion may have occurred repeatedly. A lowered water table is documented by numerous tree stumps with downward roots. These roots often reach 20 to 40 cm below the root plate, and include a thick central root going straight down (Fig. 9). A second type of root system, being spread out flat and without downward rooting to speak of, is also common at the site however. Such roots indicate a higher water table, limiting rooting space at the time of growth. This shows, that not all of the trees embedded by the expansing raised bog grew under drier conditions with significantly lowered water tables.

## 4.2 Die-off phases

Most indicative for periods of lateral bog growth are those of tree die-off. As there are seven dendrochronological dated chronology segments, there are at least seven such periods (Fig. 4). Particularly for the chronology segments A2 and B, phases of accelerated and decelerated bog advance appear to be depicted in the trees. Even though this could partially also relate to the irregular sampling pattern, some prominent short-term events of cumulative tree die-off, like those around 6315 BC, 5060 BC and 3838 BC, are clear. The periods in which the trees died off are: 6614 – 6436 BC, 6365 – 5733 BC, 5308 –





4902 BC, 4712 – 4537 BC, 3895 – 3838 BC, 3691 – 3614 BC, 3496 – 3473 BC and 3407 – 3403 BC (Fig. 4, Table 1). These are interpreted as phases of raised bog expansion. How the trees are successively affected by the local raised bog development is also illustrated in the individual tree ring series, as shown for segment F (Fig. 10). These phases of raised bog expansion should occur under relatively humid climatic conditions.

The floating chronology segment C14 shows subsequent die-off approximately. The trees of segment C14 are less well preserved. This might indicate a mire environment at the time with higher microbial activity, possibly with less continuous water logging of the mire surface.

## 4.3 Rejuvenation pulses and suppression

Several times in the record, events of numerous synchronous germination events appear. These are found following a phase
(mostly of one to a few decades) of suppressed rejuvenation. Hence, they most likely mark the end of whatever unfavourable condition suppressed rejuvenation. The increased numbers of seedling establishment can be explained by the lack of shading undergrowth as a preceding generation of young trees is missing.

Such events of bunched tree establishment (germination phases) sometimes coincide with die-off events, and more often follow directly after them. Eckstein et al. (2010, 2012) and Leuschner at al. (2007) have referred to this as Germination-
Dying-Off-phases (GDOs). Even though spatial pilot study (Stenzel, 2013) has shown, that the newly established trees are not found directly in the area shaded by those trees which had just died off, the mechanism causing the coherence is still suspected to be driven by competition for light. Other trees (e.g. *Betula*) and shrub would have suffered from the same influence killing the large pine trees and thereby created an opening in canopy. This explains the coherence or close succession of die-off and germination phases.

In the course of die-off phases, rejuvenation is often missing. When all trees of stand are killed by a fatal influence, young trees germinated relative shortly prior to the event would not occur in the record due to a lack of sufficient rings for dating. This can not be more than part of an explanation however, since the die-off phases are stretched longer in many cases than the lengths of time required of an tree ring sequence for dating. Hence, it can be suspected, that the influence killing the large
trees also affected seedling establishment.

Zackrisson et al. (1995) also take **seed production** into account to explain rejuvenation pulses in pine populations. Conditions stressful to the trees can enhance seed production. On the contrary, favourable climate conditions may also influence seed production. Enhanced seed production under stressful condition ensures rejuvenation when the older tree
generation is about to die off. The establishment of these seedlings however, depends on the opening of canopy created by the tree die-off, and on otherwise suitable conditions at the site. In how far seed production is reflected in tree establishment at the given site is therefore unclear.





### 4.4 Chronology-gaps

The site chronology of TOMO_south is interrupted by 5 gaps, ranging between 550 and 6 years in length (Table 1).

The two very short gaps (6 and 11 years respectively) are rather insignificant, especially, since the respective period is not well replicated. These gaps are likely to result from sampling design, as the neighbouring chronology segments are poorly replicated and the according trees found to the rim of the investigated area. It is well possible, that more trees represent that period, but were located just outside the margin of the site to the north-west. Both short gaps belong to the 4$^{th}$ millennium BC {3837-3832 BC (6 years) and 4613-3603 BC (11 years)} (Table 1), when trees only grew at the site on sandy elevations, forming wooded 'islands' in the mire (Fig. 5, 6 and 7).

The three longer gaps are more meaningful. The two very long gaps are 5732 - 5405 BC (328 years) and 4536 - 3987 BC (550 years), the gap of intermediate length (64 years) is 4901 – 4836 BC. All three are framed by well-replicated site chronology segments, with the respective trees found well within the site. Given the dense sampling of the site, it can be assumed, that the large gaps actually represent periods with no or very few pine trees being embedded at the site.

Other studies have attributed chronology gaps in raised bogs to a lack of tree growth due to high surface wetness. This makes much sense for the tree layers well within *Sphagnum* peat they describe, e.g. in Scotland (Moir et al., 2010) and south Sweden (Edvardsson et al., 2014). At those sites, tree growth was only possible on the raised bog during drier phases.

In case of the present study however, the stratigraphic position of the tree layer is at the fen-bog transition. Apparently, at *TOMO_south* the expanding raised bog embedded trees grown at its rim. The site was thoroughly covered with pine stumps. These were representatively sampled and dated. Taken together, this suggest, that the large chronology gaps at *TOMO_south* represent periods of (near) stagnation of the lateral raised bog growth. Therefore the gaps are not interpreted as periods of particularly high surface wetness in this case, but quite the contrary. During the periods of site chronology gaps the raised bog at *TOMO_south* appears not to have expanded significantly. This is likely related to drier periods.

### 4.5 Spatial distribution

The mapping of the dated in situ trees at the site (Fig. 5) showed the raised bog expansion from south-east to north-west. The mineral base is lowest in the south-east of the plot and highest to the south-west of the plot, as well as in the centre of the site (Fig. 6). This direction is parallel to the nearest lake shore.

There are clear spatial clusters of trees from the same periods, which usually border to patches of the preceding and following chronology segments. This is coherent with the picture of the advancing raised bog successively embedding the tree stumps.

The oldest parts (with trees only from the first chronology segments) are found to the south-east, where the mineral base is low. The trees from more recent centuries were found in places were the mineral base ascends, like the sandy elevation near the centre of the plot. Apparently wooded islands within the raised bog persisted for a long time (Fig. 5, 6 and 7). The





distribution of trees from different chronology segments in the plot can mostly be related to the dynamic relief of the mineral base (Fig. 6).

At some places however, there are also trees from different epochs interspersed (Fig. 5). In the mapping of raised bog expansion (Fig. 5) these places are assigned to the last tree die-off they document, for the advancing central part of the bog should have been generally treeless (Ellenberg, 1996). Re-establishment of trees in places were trees had already been embedded before may have been favoured by drier conditions, or simply relate do the dynamic relief, with the root plates of previous generations additionally serving as 'stepping stones' for the new trees to grow on.

### 4.6 Peat stratigraphy

Thick and mostly weakly humified layers of brown moss peat were found at the bottom of *TOMO_south* peat cores. It was the base for extended tree growth, which then itself deposited thick layers of wood rich peat. Were basal swamp forest peat is less humified, the brown moss can still be seen intertwined. Likewise intertwined is the first *Sphagnum* growth into the top layers of pine rich peat. This depicts a succession to ombrotrophy, with one plant community creating the habitat for the next.

There are fine mineral materials found interspersed in several peat cores and layers. These may have been washed into the moss by temporary flooding or, particularly in the older deposits, also might have been blown into the mire.

### 4.7 Climatic comparisons

The following alignments are a few examples only. Particularly such records were picked, which have a certain precision of dating (dendrochronological, varve, etc.), and are located with some level of proximity (European studies). Furthermore, studies making statements on temperature only were mostly neglected, as the pines of *TOMO_south* are taken to be affected predominantly by changes of humidity. For the type of proxy used in the studies referred to and their location, please see Table 2. The temporal overview given in the following text is also displayed in Figure 11.

We interpret the **die-off phases** as times of lateral mire expansion. Bog expansion again would require a certain humidity, which may be more dependent on a reduced evapotranspiration than the actual precipitation alone. This would mean the die-off phases in the periods c. 6600– 6450 BC, 6350 – 5750 BC, 5300 – 4900 BC, 4700 – 4550 BC, 3900 – 3850 BC, 3700 – 3600 BC, 3500 – 3400 BC indicate more humid phases. In turn, the **gaps** in the site chronology and the periods of rather undisturbed tree growth are interpreted as phases of stagnation of lateral raised bog expansion. These may be related to climatic conditions unfavourable for bog growths, possibly involving drier periods. This would apply to the phases c. 6450 - 6350 BC, – 5750 - 5300 BC, 4900 - 4700 BC, 4550 - 3900 BC, 3850 - 3700 BC, and 3600 - 3500 BC.

As mentioned before, the die-off phases observed at *TOMO_south* show much synchronism with die-off phases observed at other mire sites in the region ( Eckstein et al., 2009, 2010, 2011), which emphasises their climatic context.





The **8.2k** event, a global cooling period, began around c. 6300 cal. BC (Thomas et al., 2007) or around 6225 cal. BC (Kobashi et al., 2007), while there appears to be evidence of it in the form of concurrent pine establishment at three bog sites in Ireland from 6210 BC on (Torbenson et al., 2015). At site *TOMO_south*, a phase of frequent tree die-off from 6250 BC to 6157 BC matches the time of maximum cooling (6250–6150 varve years BC) as described by Veski et al. (2004).

The die-off phase of site chronology segment A2 (c. 6350 – 5750 BC) covers the time of the event and might be partially influenced by it. The trees display a strong die-off pulse at c. 6315 BC, which might be related to the beginning of the 8.2k cooling period, or predates it.

The pines of *TOMO_south* do not show a reaction as pronounced to the 8.2k event as found in other German dendrochronological records. The west German oaks from the river Main show poor growth and regeneration conditions c.
6270 – 6000 BC (Spurk et al., 2002). The northwest German bog oak chronology even has a gap 6177 – 6060 BC (Achterberg et al., 2016), the end of which fits the varve-inferred abrupt end of the 8.2k cooling period described for 6080 varve years BC by Veski et al. (2004).

The 8.2 k event cooling phase, according to Thomas et al. (2007), began around c. 6300 cal. BC. Kobashi et al. (2007) date its beginning later, to c. 6225 cal. BC, while Veski et al. (2004) observe the time of maximum cooling for 6250-6150 varve years BC. Veski et al. (2004) also observe an abrupt end of the 8.2 cool period for 6080 varve years BC. Dendrochronological data in proposed context with the event includes a phase of pine establishment in Irish bogs from 6210 BC on (Torbenson et al., 2015), a phase of poor growth and regeneration of west German oaks from the river Main
sediments c. 6270 – 6000 BC (Spurk et al., 2002) and a gap in the northwest German bog oak chronology 6177 – 6060 BC (Achterberg et al., 2016).

These dates largely fit together, even though they also show some divergence. Particularly the end of the cool phase stated by Veski et al. for 6080 varve BC and the end of the bog oak chronology gap 6060 BC correspond nicely. The pines of northwest Germany, including those of *TOMO_south*, display a prominent die-off pulse c. 6315 BC, which might predate the
beginning of the 8.2k cooling period rather than being related to the event. The whole scope of 8.2k related dates is contemporary to a rather well replicated section of the pine chronology, covered by the site chronology die-off phase of segment A2 (c. 6350 – 5750 BC). The pines do not show any strong reaction clearly related to the event.

Magny at al. (2004) describe several phases of high lake levels in mid-Europe, four of which are within the time frame covered at *TOMO_south*. The first of these (c. 6350-6100 cal. BC) is also within the die-off phase of segment A2 (6365 –
5733 BC). The beginning of the two phases (high lake levels in Magny et al. 2004 and die-off at *TOMO_south*) are in close temporal accordance. The second phase of high lake levels, described by Magny et al. (2004) for c. 5600 – 5300 cal. BC does not fit the data of *TOMO_south.* Its beginning is contemporaneous to a site chronology gap at *TOMO_south*, which is taken to indicate dry conditions rather than wet ones, and its end is contemporaneous to the beginning of a die-off phase at *TOMO_south* (segment B, 5308 BC), which should indicate the begin of a more humid period rather than its end.





The data of Schmidt at al. (2004) seems to be contradicting the record of *TOMO_south* at that time as well. They show data from 5600 BC on, which displays more or less humid conditions until about 5420 BC. This is within a chronology gap at *TOMO_south*, with the gap end (5404 BC) closely meeting the end of the relatively humid conditions observed in the data of Schmidt et al.(2004). The subsequent dry phase displayed by Schmidt at al. (2004) for c. 5400 - 5350 BC fits the indications from *TOMO_south* better, where the pines begin establishment contemporary to the dry phase beginning, and start dying off only after the dry phases end, in 5308 BC. The two wet phases that follow according to Schmidt et al. (2004) (c. 5320-5000 BC and c. 4950-4900 BC) fit the die-off phase of segment B at *TOMO_south* (5308-4902 BC), which also indicates humid conditions. The beginning of the above die-off phase is temporally close to the beginning of the first of the two mentioned wet phases, and the end of the die-off phase is contemporary to the end of the second. This is a very close agreement of the indications of the two dendrochronological records. The interjacent dry phase (c. 5000-4950 BC) documented by Schmidt et al. (2004) is not reflected in *TOMO_south*.

Gunnarson et al. (2003) describe drier conditions for c. 4900-4800 BC. At *TOMO_south* a site chronology gap (4901-4838 BC) begins at the same time. The gap and the following phase of tree establishment comply with drier conditions.

More humid conditions are evident from pollen and peat data composed in context of a trackway (Bauerochse 2003), which is dendrochronologically dated (construction and maintenance 4629 - 4545 BC) (Bauerochse et al. 2012, Achterberg et al. 2015). The palaeo-botanical indications for increased humidity described by Baueroche (2003) can thus be aligned to the die-off phase of site chronology segment C (4712-4537 BC).

The third of Magny et al.'s (2004) phases of high lake levels (c. 4400-3950 cal. BC) does not coincide with indications for increased humidity at *TOMO_south*. It begins within a long site chronology gap (4536 - 3987 BC) and ends before the beginning of the next die-off phase. Turney at al. (2006) on the other hand state drier conditions for Ireland around c. 4250 BC, also dating within the *TOMO_south* site chronology gap 4536 - 3987 BC, which is in better agreement with our interpretation of the *TOMO_south* data.

Palynological indications for increased humidity (Bauerochse 2003) are temporally anchored to after 3701 BC by a dendrochronologically dated trackway (Bauerochse et al. 2012, Achterberg et al. 2015). This is within the die-off phase of segment E at *TOMO_south* (3691-3614 BC), and thereby i n agreement with its climatic indication. Around the same time Arbogast et al. (2006) as well identify climatic deterioration, a shift towards cooler and possibly more humid conditions for c. 3700-3250 BC. Magny at al. (2004) date the beginning of their fourth phase of high lake levels to c. 3700 cal. BC as well (c. 3700-3250 cal. BC). Despite the low replication of segment E, these coherences make the indication of its die-off phase appear quite valid.

The wet phase observed by Gunnarson et al. (2003) for c. 3600 – 3400 BC covers about the same time as segment F. This is not exactly analogue. However, both die-off pulses of segment F thereby are within the wet phase documented for Sweden (Gunnarson et al. 2003), with the end of the wet phase meeting the end of the last die-off pulse. This makes for an intermediate level of agreement.



Dreslerova (2012) points out a pronounced shift towards wetter, cooler and a more variable climate around c. 3550 cal. BC, reviewing numerous European studies of climate proxy for the Holocene. This pre-dates the die-off phase 3496-3473 BC (segment F) by half a century, but may well be related, as Dreslerova (2012) describes this date to be the beginning of a climatic phase.

In general, there is much agreement with other climatic records, but also some divergence. This is plausible, seeing that the mire development reflects climatic conditions on the one hand, but on the other hand represents a local signal.

## 5 Conclusions

The trees, stratigraphically located at the fen-bog-transition, are viewed to stem from the former bog margin, being embedded by the expanding raised bogs *Sphagnum* peat. The tree dates document a raised bog expansion at the site for 6614 – 6436 BC (c. 250y), 6365 - 5733 BC (c. 750y) , 5308 – 4902 BC (c. 500y), 4712-4537 BC (c. 300y), 3895-3838 BC (c. 150y), 3691-3614 BC (c. 200y), 3496-3473 BC (c. 25y) and c. 3400 BC (4y). These phases of lateral raised bog growth likely occurred in periods of rather humid climate.

The shorter gaps in the later part of the site chronology are viewed insignificant, since they belong to poorly replicated periods documented at the sites margin. The three longer gaps in the earlier part of the site chronology however are framed by well replicated sections represented in central area of the site. These are interpreted to represent periods without significant (and lasting) raised bog advance, since the site is throughout covered with stumps and these were densely sampled. These chronology gaps relating to phases of apparent bog stagnation are 5733- 5308 BC (328 y), 4902 -4712 BC (64y) and 4537 – 3895 BC (550y). They likely connect do drier climatic phases.

The distribution of the tree stumps of various ages across the site supports the picture of subsequent bog advance embedding the tree stumps. The bog expanded from the south-east towards north-west according to the dating of the trees, the direction of bog growth largely reflecting the elevation of the mineral base below the peat. Rises of the sandy ground formed wooded islands within the bog, being successively covered by *Sphagnum* peat with much delay.

## 6 Acknowledgements

Thanks goes to Torsten Struck for support and generous help in field work, Jens Will for a final language check-up, and all reviewers and editors. We would particularly like to express our gratitude to ASB-Humus, especially Mr. Thuernau, for kindly allowing us to work on their peat extraction grounds, their helpful support of our research and for sharing their elevation data. Our sincerest gratitude goes to the German research society DFG for funding the work this manuscript is based on (projects LE 1805/2 and HA 4438/1).



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



**Table 1:** Chronology cover. For this table, the measured rings were used only, estimations of missing rings to pith or bark were not added. Only the trees with dendrochronological calendar dates are listed below.

| Chronology segment name | A | | B | C | D | E | F | total |
|---|---|---|---|---|---|---|---|---|
| | **A1** | **A2** | | | | | | |
| Color | Dark violet | Blue | Pink | Red | Orange | Yellow | Green | |
| Chronology cover [y BC] | 6703 – 5733 | | 5404 – 4902 | 4837 – 4537 | 3986 – 3838 | 3831 – 3614 | 3602 – 3403 | Dispersed over 3301 years |
| | 6703 – 6436 | 6483 – 5733 | | | | | | |
| Tree die-off [y BC] | 6614 – 6436 | 6365 – 5733 | 5308 – 4902 | 4712 – 4537 | 3895 – 3838 | 3691 – 3614 | 3496 – 3473 3407 – 3403 | 1482 years |
| Chronology segment length [y] | 971 | | 504 | 301 | 149 | 218 | 202 | 2345 years |
| | 268 | 752 | | | | | | |
| Chronology gap [y] | - | 328 | 64 | 550 | 6 | 11 | - | 959 years |
| Replication [trees] | 254 | | 66 | 123 | 10 | 19 | 5 | 477 trees |
| | 53 | 201 | | | | | | |

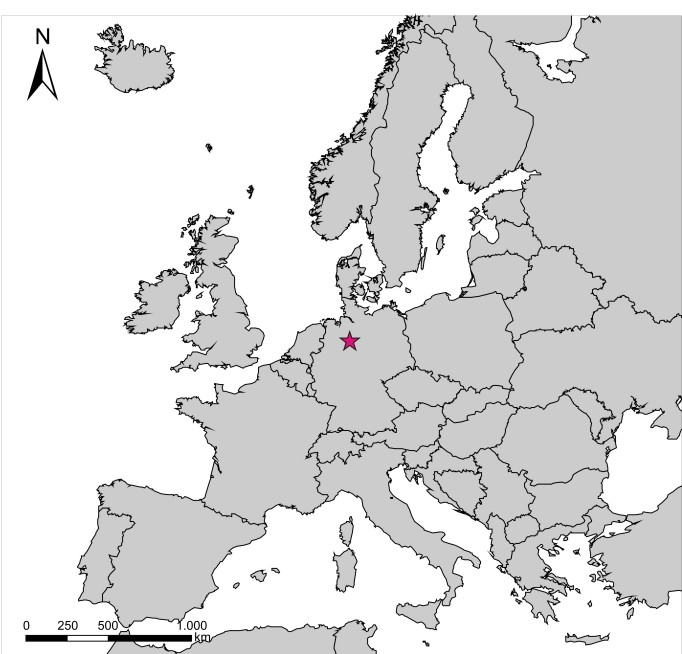

**Figure 1:** Location of the study site *TOMO_south*, indicated by a star.



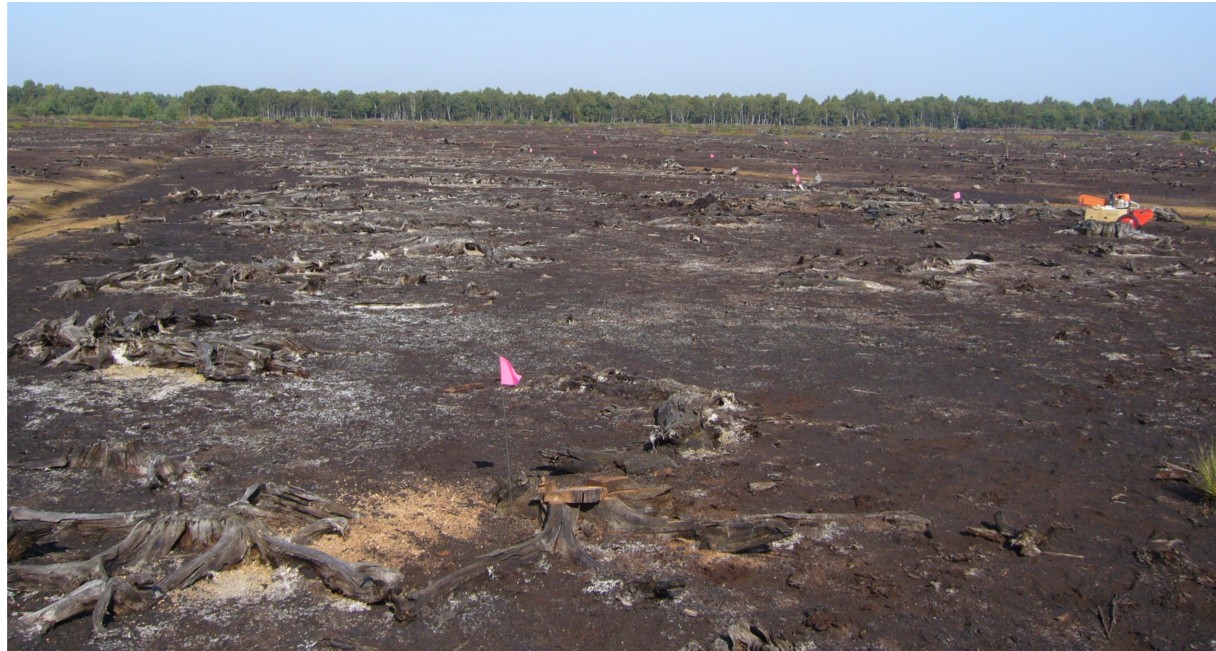

**Figure 2:** View on site *TOMO_south*. The upper layers of peat have been removed. Numerous tree-stumps are protruding. Photo: Inke Achterberg.

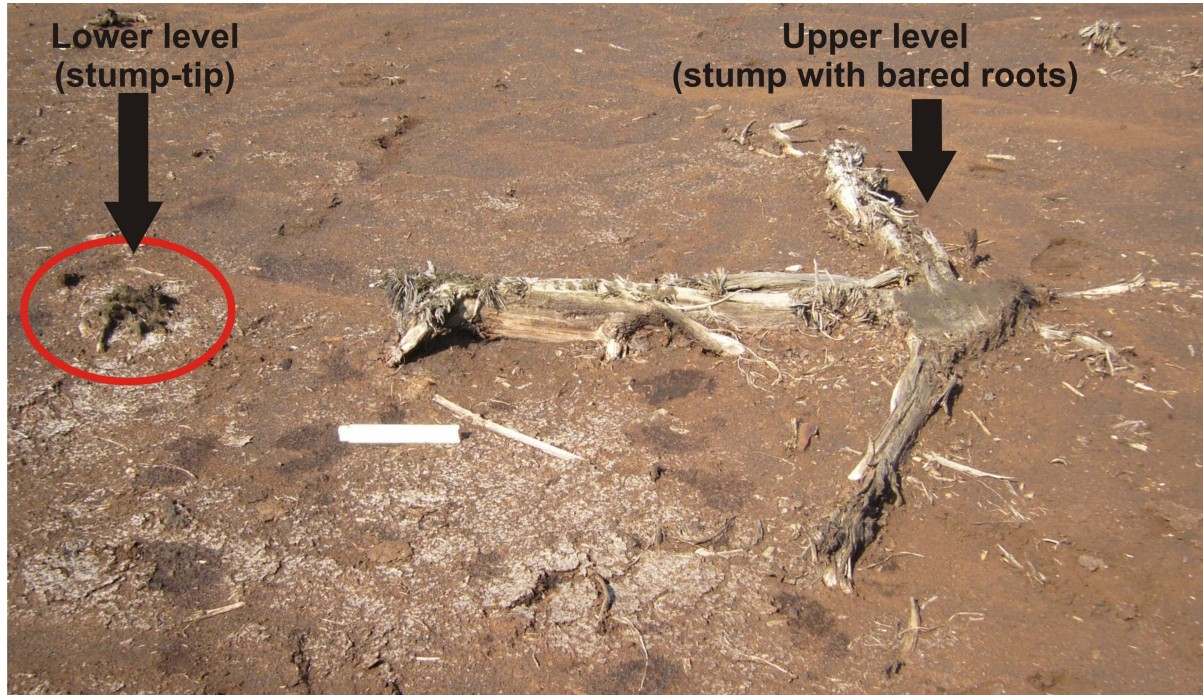

**Figure 3:** Tree stumps at site *TOMO_south* grown on two elevation levels. The upper level trees are sometimes supported by their dead successors. Photo: Inke Achterberg.







**Figure 4:** Temporal distribution of pine trees at site *TOMO_south*. On top the chronology segments C14 and A1-F are indicated. The gaps in the site chronology (white) are labelled with their duration. The time covered by the site chronology of *TOMO_south* is underlain in grey, for the floating chronology segment C14 striped in grey-white. The coloured horizontal lines indicate the lifespans of the individual dated trees from the site (measured rings coloured, estimated missing rings black). The trees of each chronology segment (Table 1) are shown in a different colour (A1: dark violet, A2: blue, B: pink, C: red, D: orange, E: yellow, F: green), the floating chronology segment C14 in grey. The colours are used accordingly in the spatial mapping (Fig. 5 and 6). The oaks from the site are displayed in black, below the pines. In [A] the trees are sorted by die-off date. The three most prominent events of cumulative tree die-off are highlighted by a dashed blue vertical line and labelled by year BC. Below, the periods from first to last tree die-off (last measured ring) are indicated, labelled by years BC. The lighter grey backgrounds indicate when the trees of a chronology segment start to die off. In [B] the trees are sorted by germination date. Here, the lighter grey backgrounds indicate periods with limited tree establishment.





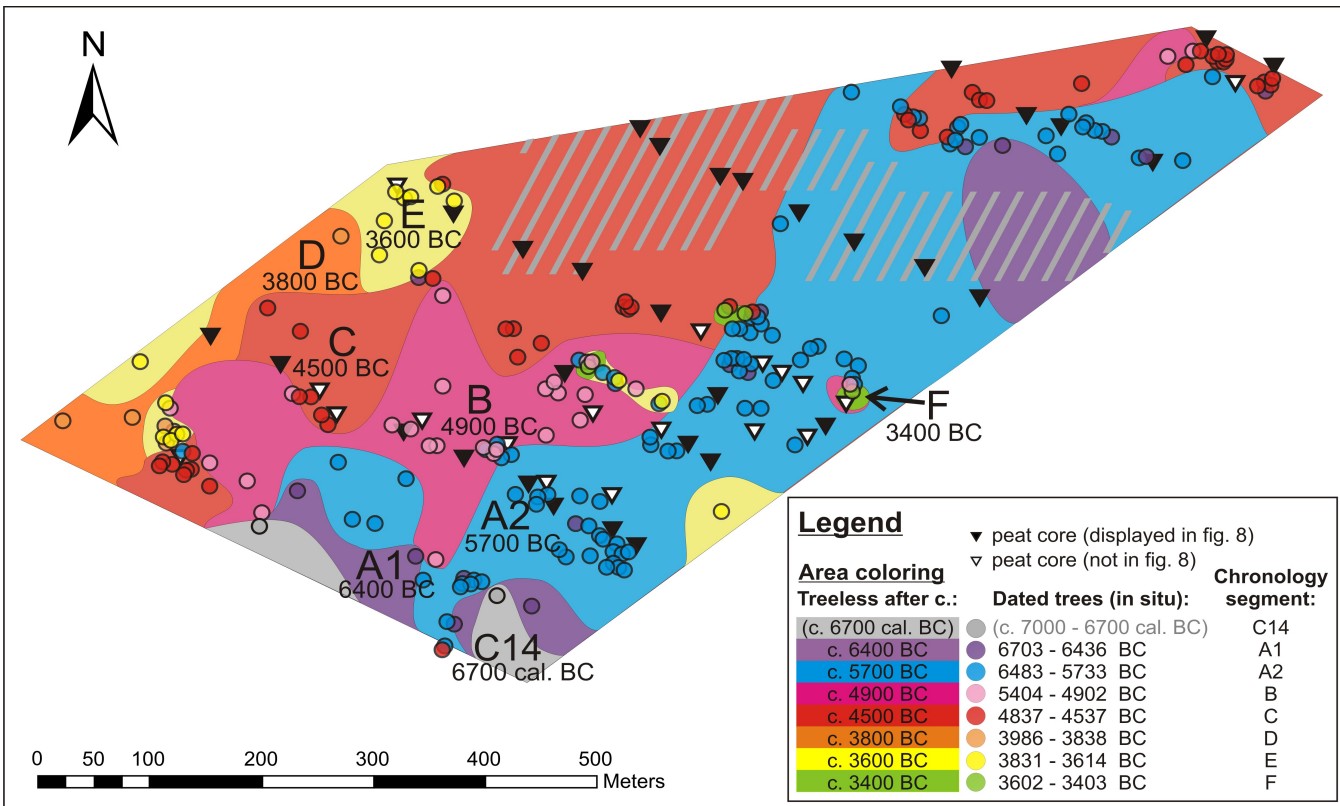

**Figure 5:** Reconstructed raised bog advance in terms of 'no more tree growth'.The coloured dots indicate in situ tree remains, their colour indicates the chronology segments they belong to. The hachures indicate areas with lac of dated in situ samples. The coloured areas show, where NO MORE trees grew at a certain period. In this sense, the oldest section is where only trees of the floating, radiocarbon dated segment C14 (beginning of the 7th millennium cal. BC) were found, which is coloured grey. The dark violet areas show, were no trees younger than the first group of trees with dendrochronological calendar dates, A1 (early 7th millennium BC), were found. Blue areas feature only trees from the late 7th and early 6th millennium BC (A2) and older. The pink area delivered trees from the second half of the 6th millenium BC and the beginning of the 5th (B). The red area shows were no trees grew after the first half of the 5th millennium BC (C). The orange indicates were, after that, only trees from 4000-3800 BC (D) were found. The yellow area still featured trees in the first half of the 4th millennium BC (E), and the green in the second half of the 4th millennium BC (F). The colours with the according dates are given to the right.





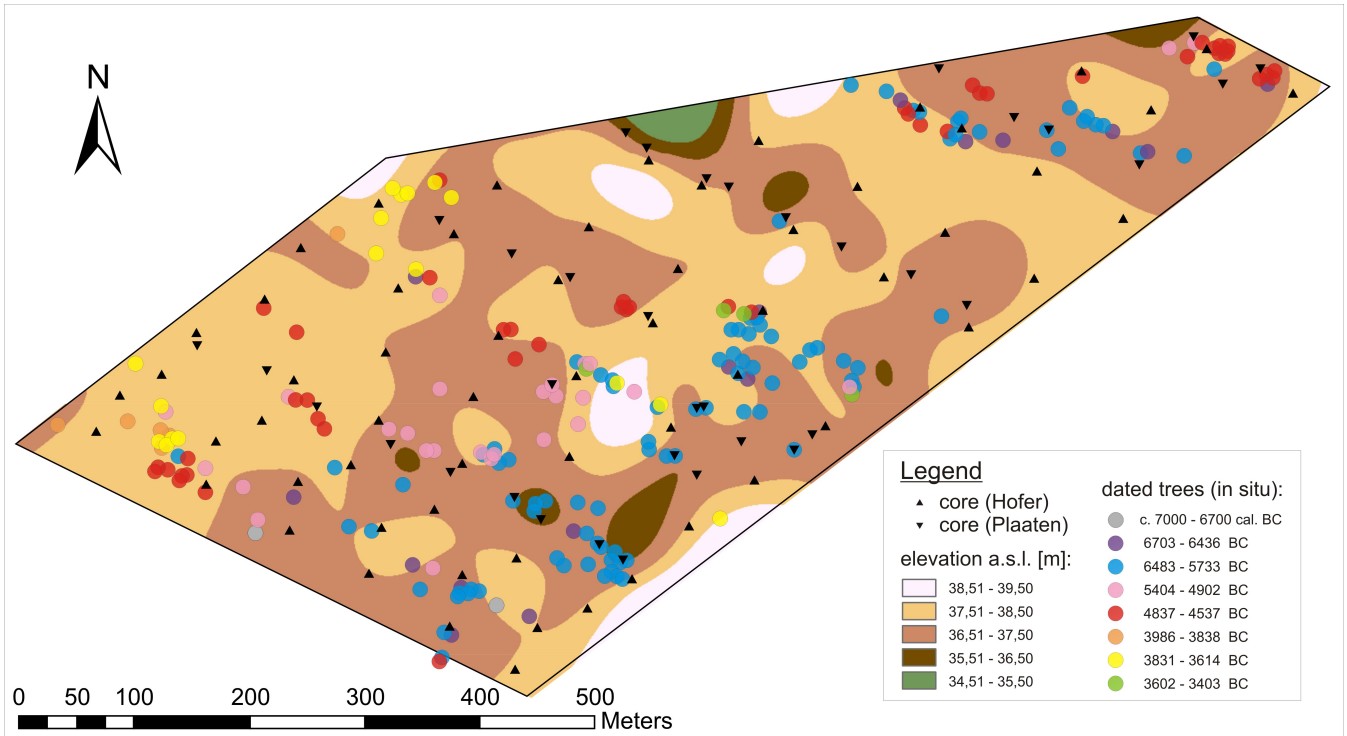

**Figure 6:** Elevations a.s.l. of the mineral base (sand) below the peat. Regularized spline interpolation of 99 measurements. The elevations are interpolated using a regularized spline. The actual measurements rage from 36,42 to 38,16 m a.s.l.; thereby, the first and the last elevation class (green, below 35,50 m a.s.l., and white, above 38,51 m a.s.l.) are products of extrapolation only. The coring points are indicated by triangles. The dated in situ trees are shown as circles, for each chronology segment (Table 1) in a different colour (C14: grey, A1: dark violet, A2: blue, B: pink, C: red, D: orange, E: yellow, F: green).

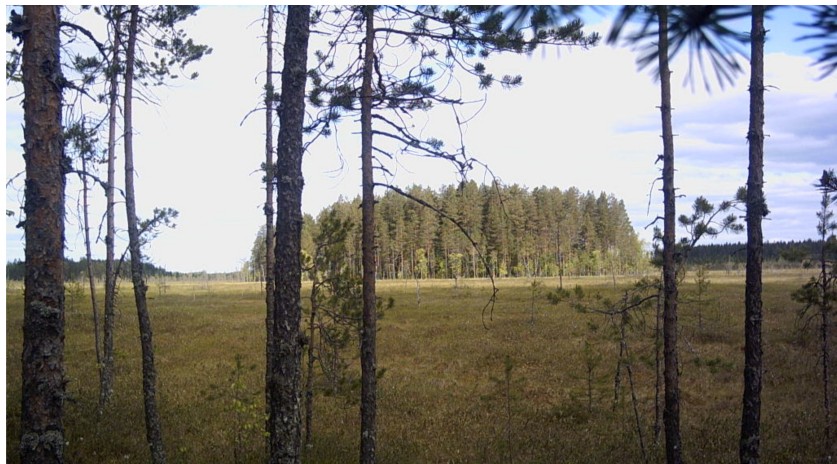

**Figure 7:** Pine growth at a bog site in Finland. View across a stretch of treeless bog. In front are small trees, thinning out where the peat deepens. In the back large trees form a wooded 'island' where the mineral ground ascends. Photo: Inke Achterberg.





**Figure 8:** Peat stratigraphy. Six transects are shown with the peat stratigraphy and elevation a.s.l. of the cores. In the sky view, site TOMO_south is outlined in pink. The locations of the displayed peat cores are indicated by black triangles, yellow lines connecting the transects. Peat stratigraphy for cores without elevation measurement is not shown. Please note, that cores D8, F1 and F2 were taken at the border to the plot, where peat was left standing significantly higher than within the plot where the tree layer was exposed. Underlying satellite image: ©2008 Google Earth, image ©2009GeoContent, ©2009 Tele Atlas.





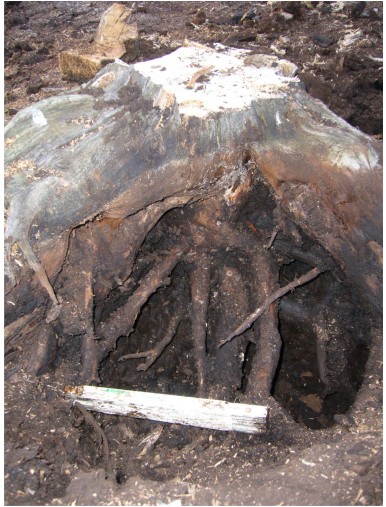

**Figure 9:** Strong downward roots below a tree stump *TOMO_south* (tree life 5396-5230 BC, chronology segment B). Photo: Inke Achterberg.

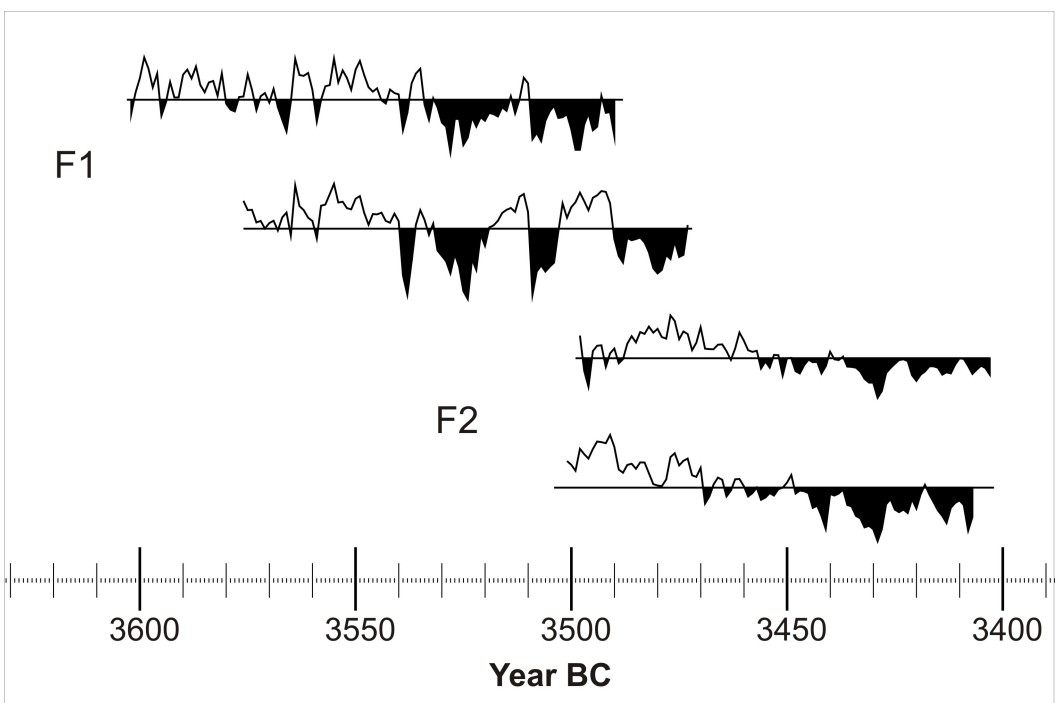

**Figure 10:** Tree ring widths curves of chronology segment F. The mean value of the tree ring widths is indicated by a horizontal line for each curve, the area below mean is filled black to highlight growth depressions. One sample is not shown, because it is missing several rings to bark and thereby the record of the final years of that trees life. All trees show narrow ring widths prior to their deaths. These growth depressions are not reflected equally in the surviving trees. This illustrates the locality of the calamity.



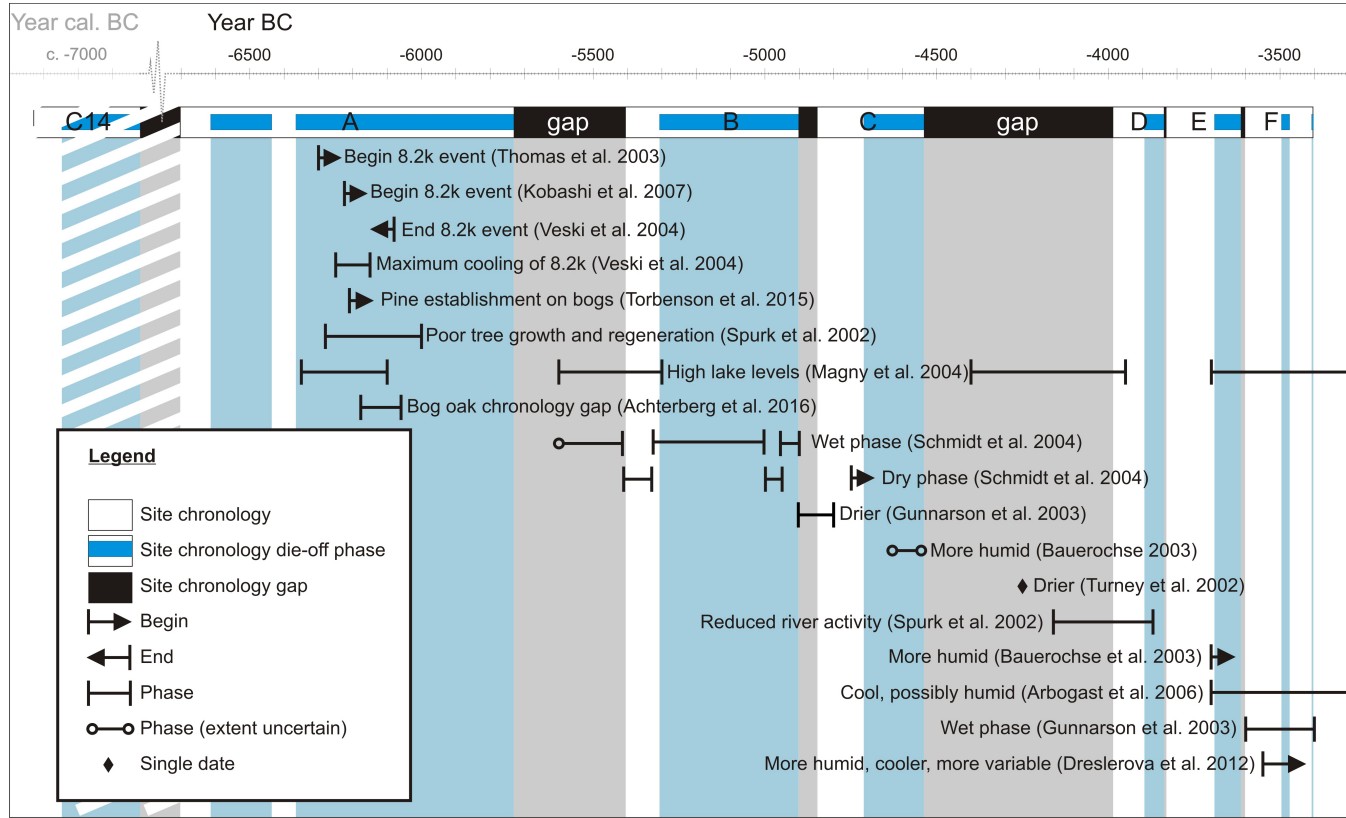

**Figure 11:** Comparison of the phases documented at *TOMO_south* with those of a few other studies.

**Table 2:** A few comparisons with other palaeoenvironmental studies, predominantly such with precise dating.

| Publication | Proxy | Location | Time | Indication | *TOMO_south* | fit |
|---|---|---|---|---|---|---|
| Thomas et al. (2007) | Ice cores chemistry and stable isotope | Greenland | Begin c. 6300 cal. BC | Cooling | Die-off phase of site chronology segment A2 (c. 6350 – 5750 BC) ; | + |
| Spurk et al. (2002) | Dendrochronological | West Germany | c. 6270 – 6000 BC | Poor growth and regeneration conditions | strong die-off pulse at c. 6315 BC | |
| Kobashi et al. (2007) | Ice core (GISP2) methane and nitrogen isotopes | Greenland | Begin c. 6225 cal. BC | Cooling | | |
| Torbenson et al. (2015) | Dendrochronological; pine establishment at three bog sites | Ireland | From 6210 BC on | | | |
| Veski et al. (2004) | Varve | Estonia | 6250 – 6150 varve years BC | Maximum cooling | Frequent tree die-off from 6250 BC to 6157 BC | + |
| Veski et al. (2004) | Varve | Estonia | 6080 varve years BC | Abrupt end of the 8.2k cooling period | | |
| Achterberg et al. (2016) | Dendrochronological | Northwest Germany | 6177 – 6060 BC | Gap in bog oak chronology | | |
| Magny (2004) | Radiocarbon, dendrochronological and | France and Switzerland | c. 6350 - 6100 cal. BC | High lake levels (I) | Start is beginning of the die-off phase of segment A2 at *TOMO_south* (6365 BC) | + |



| | | | | | |
|---|---|---|---|---|---|
| | | | archaeological dates | | |
| Magny (2004) | Radiocarbon, dendrochronological and archaeological dates | France and Switzerland | c. 5600 - 5300 cal. BC | High lake levels (II) | End coincides with beginning of die-off phase (5308 - 4902 BC) | +/- |
| Schmidt et al. (2004) | Dendrochronological | West Germany | (5600) - 5420 BC | More or less humid | Long gap (5732 - 5405 BC) | - |
| Schmidt et al. (2004) | Dendrochronological | West Germany | c. 5410 - 5330 BC | Dry phase | Tree establishment and growth (no die-off) | + |
| Schmidt et al. (2004) | Dendrochronological | West Germany | c. 5320 - 5000 BC | Wet phase | Beginning at the same time as die-off phase c. 5300 – 4900 BC begins | + |
| Schmidt et al. (2004) | Dendrochronological | West Germany | c. 5000 - 4950 BC | Dry phase | (Not reflected clearly in the data of *TOMO_south*) | - |
| Schmidt et al. (2004) | Dendrochronological | West Germany | c. 4950 - 4900 BC | Wet phase | End at the same time as die-off phase c. 5300 – 4900 BC ends | + |
| Gunnarson et al. (2003) | Dendrochronological | Sweden | c. 4900 - 4800 BC | Drier conditions | Found accordingly for *TOMO_south* (c. 4900 - 4700 BC) | + |
| Bauerochse 2003 | Palynological implications, context of dendro-dated find | NW Germany | Context of find dated to 4629 -4545 BC | More humid conditions | Die-off phase 4712 - 4537 BC | + |
| Magny (2004) | Radiocarbon, dendrochronological and archaeological dates | France and Switzerland | c. 4400 - 3950 cal. BC | High lake levels (III) | Gap in the site chronology (c. 4500 - 3900 BC) | - |
| Turney at al. (2006) | Dendrochronological | Ireland | c. 4250 BC | Drier conditions | Within the gap of the site chronology 4536 - 3987 BC | + |
| Spurk et al. (2002) | Dendrochronological | West Germany | c. 4160 - 3870 BC | Reduced activity of the river Main | Gap 4536 - 3987 BC, which might point towards drier conditions as well. Die-off phase at *TOMO_south* from 3895 BC on, 25 years prior to the end of the period identified by Spurk et al. (2002). | +/- |
| Bauerochse (2003) | Palynological implications, context of dendro-dated find | NW Germany | after 3701 BC | More humid conditions | Die-off phase at *TOMO_south* from 3691 BC to 3614 BC | + |
| Arbogast et al. (2006) | Dendrochronological dates from archaeological layers | Swiss Alps, French and German | 3700 - 3250 BC | Lake level rise; climatic deterioration, cool and possibly humid conditions | Beginning die-off phase segment E (3691 BC), (covers die-off phases segments E and F) | + |
| Magny (2004) | Radiocarbon, dendrochronological and archaeological dates | France and Switzerland | c. 3700 - 3250 cal. BC, maximum 3350 - 3250 cal. BC | High lake levels (IV) | Start is beginning of die-off phase segment E (3691 BC) | + |
| Dreslerova (2012) | Multi-proxy (review) | Europe (multi site) | c. 3550 cal. BC | Pronounced shift towards wetter, cooler and a more variable climate | Shift from stagnating bog growth to a phase of bog expansion found for c. 3500 BC | + |
| Gunnarson et al. (2003) | Dendrochronological | Sweden | c. 3600 – 3400 BC | Wet phase | Contemporary to cover of segment F (3602-3403 BC), including two die-off pulses | +/- |
| Magny & Haas (2004) | Dendrochronological date, pollen and archaeological | Switzerland | c. 3370 BC | Lake level rise | Last trees at *TOMO_south* died off 35 years previously | - |