# Peer review of "Dendrochronologically dated pine stumps document phase wise bog expansion at a northwest German site between c. 6700 BC and c. 3400 BC"

_Climate of the Past, 2017_

## Referee Comment (RC1) · Anonymous Referee #1 · 13 Apr 2017

The paper by Achterberg et al focuses on the reconstruction of the temporal development of a mire/bog system in northwest Germany, using dendrochronologically dates Pine stumps preserved in the peat. I will comment the aspects relative to peat development/dynamic, and the implications for the Holocene climate history, leaving to more dendro specialists a more insightful assessment of the accuracy and scope of the crossdating approach.

This study produces a nice dataset of dendrochronologically dated pine stumps, and allow assessing the temporal and spatial evolution of the bog, together with its relative

drivers (mainly hydrological changes), tracked by distinct episodes of tree mortality.

In general I found the paper difficult to follow in many parts, and I suggest major restructuring (see below) and a more careful editing by an English native speaker. Just to mention few examples: "In situ finds were later separated from ex situ finds" (P. 3), "A second type of root system, being spread out flat and without downward rooting to speak of, is also common at the site however" (P.6), "The following alignments are a few examples only" (P. 9). I have difficulty to understand what the authors mean in these and other parts of the text. In addition, many statements seem not to be supported by data evidence, or not fully justified, as for e.g. "which may be more dependent on a reduced evapotranspiration than the actual precipitation alone". (P. 9). Why is this the case?

The implications of this study for our understanding of past climate are also rather unclear. The discussion on past climate changes seems to be quite general, and too marginal compared to the aspects of peat development. Most importantly, the authors get to the conclusion that "the mire development reflects climate conditions on the one hand, but on the other hand represents a local signal" (P12), but no discussion about land-use changes in the area is provided. Therefore, the extent to which tree mortality truly reflect changes in moisture remain unclear, considering that data cover a period of time (the Neolithic) where deforestation in the catchment may have resulted in a raise of the water table, thus biasing the climate signal.

In sum, I value the big effort from the authors to provide such a nice dataset, by I suggest a major revision and improvement in most chapters, to better fit the Journal's scope.

Sorry not to be more helpful at this stage.
* * *

---

## Referee Comment (RC2) · Anonymous Referee #2 · 18 Apr 2017

Review of: Title: Dendrochronologically dated pine stumps document phase wise bog expansion at a northwest German site between c. 6700 BC and c. 3400 BC Author(s): Inke Elisabeth Maike Achterberg et al. MS No.: cp-2017-4 MS Type: Research article

This appears to be a well designed and well executed study that is of importance to a better understanding of long-term climate-hydrology-vegetation dynamics in peatlands. I think the study has the importance and quality to be published in Climate of the Past. Before publication, however, several questions have to be addressed and some changes in the text and figures are needed to improve the paper.

I have three significant criticisms of the paper: 1. I think that one of the unique and important aspects of the paper is the usage of dated roots reaching different levels in the peat bog. The usage of horizontal versus vertical roots makes it possible to precisely date water table lowering in the peat bog. Moreover, the usage of dated vertical roots in combination with tree-ring width data provides a possibility to distinguish between water table rise and lowering, which both have the potential to cause radial tree-growth depressions. The information from the tree growth and the roots combined, however, should therefore be something that is used and displayed much stronger in the paper. The usage of the roots, and the associated information is for example not even mentioned in the abstract. 2. In general, the paper is well-written, though with some word choices and syntax that likely reflect English not being the authors' first language. I have therefore made some editorial suggestions along those lines. But, neither I have English as first language, I therefore suggest that the paper needs to be proofread by a native English speaker, preferably with knowledge in written scientific language, to improve the grammar and flow of the paper. I think this would improve the overall impression of the paper a lot. 3. In the end of the discussion (page 12, line 5-6), it's written that "In general, there is much agreement with other climate records, but also divergence. This is plausible, seeing that the mire development reflects climatic conditions on the one hand, but on the other hand represents a local signal." Does it mean that the data is okay when it correspond to other records, and that the local signal blurs the climatic influence when it doesn't fit other studies? If so, we don't learn much new things from the study. Moreover, if we don't have other records to compare to, what parts of your data series can we trust and what parts are just reflecting local mire development? I think that you need to explain during what conditions the new data presented is better than other records, and during what type of conditions there is a disadvantage compared to other records. Maybe the usage of roots and tree-ring data combined can make it possible to distinguish between different types of hydrological changes (wet-shifts and droughts) in a more accurate way than other studies. If so, there is great potential with the data, interpretations and the manuscript.

[Figure]

ABSTRACT As already mentioned, the usage of the root depth and the associated information is absent in the abstract. This is something important and unique that should be stressed much more in the paper. The design of the abstract can also be improved if the results of the study are highlighted instead of a description of the methods. Page 1, line 11: In the third sentence, it's written that it's a "dated site chronology", but I suggest that it should be changed to "five dated site chronologies" as there are gaps between them. Page 1, line 14: change "larger period" to "longer period"

INTRODUCTION Page 2, line 2: Is it Pinus sylvestris or Pinus spp. ? Page 2, line 3: I suggest that "The expansion of the raised bog often killed trees" to "The expansion of raised bogs during moist periods often cause severe growth conditions for bog trees and consequently widespread dying-off phases.

Page 2, line 9: There is a review paper about bog trees that I suggest being mentioned here: Edvardsson, J., Stoffel, M., Corona, C., Bragazza, L., Leuschner, H.H., Charman, D.J., Helama, S. 2016. Subfossil peatland trees as proxies for palaeohydrology and climate reconstruction during the Holocene. Earth-Science Reviews 163, 118-140. The following study from Poland might also be relevant to mention: Krapiec, M., Margielewski, W., Korzen, K., Szychowska-Krapiec, E., Nalepka, D., & Lajczak, A. (2016). Late Holocene palaeoclimate variability: The significance of bog pine dendrochronology related to peat stratigraphy. The Puscizna Wielka raised bog case study (Orawa-Nowy Targ Basin, Polish Inner Carpathians). Quaternary Science Reviews, Volume 148, p. 192-208., 148, 192-208.

Moreover, I think that several results and studies presented in this review paper by Edvardsson et al. (2016) would be useful and improve the discussion when comparisons to other studies are made later in the manuscript (pages 9 to 11).

Page 2, line 13: The paper Edvardsson et al., 2011 was published 2012: Edvardsson, J., Leuschner, H.H., Linderson, H., Linderholm, H.W., Hammarlund, D. 2012. South

Swedish bog pines as indicators of Mid-Holocene climate variability: Dendrochronologia 30, 93-103.

MATERIAL AND METHODS

Page 2, line 20-23: I think the first sentences in the section can be improved. Suggestion, "The Tote Moor mire complex is located north of lake Steinhuder Meer near Hanover. The undulating relief bellow the mire consist of sand, and is likely to have held several small ponds and isolated mires before the expanding mire complex connected them". Use the word "sand" only if it's "sand", otherwise e.g. "minerogenic material", "mineral soil", or "glaciofluvial depiosits" might be better.

Page 2, line 29: It should be "0.01 mm" with a dot, not comma.

Page 2, line 30: It should be "TSAPWin" (TSAP with capital letters).

Page 3, line 4: In situ is sometimes written in italics and sometimes not. I should be consistent according to the guidelines of Climate of the Past.

Page 3, line 6: I think "About 96 trees have possibly been moved. . .." reads better."

Page 3, line 6: What is "in situ s.s. finds"?

RESULTS

Page 3, line 25: How many pine stumps does "many pine stumps" represent?

Page 5, line 20: I think that "first" and "second" group would read better. For example, " the first type of root system (type 1) spreads horizontally without any downward pointing roots. The central root at these trees has either died off at a length of about 10 cm or less, or is not traceable at all. The second type of root systems (type 2), however, displays downward growing roots, most often with a pronounced central root that has grown vertically downwards". Maybe "vertical" and "horizontal roots" are better than "flat" and "downward roots".

[Figure]

Page 5, line 25: Maybe "mineral soil" is better and more accurate to use than "sand".

DISCUSSION Page 6, line 4: "at the raised bog margin"

Page 6, line 10: "peat stratigraphical" (two words)

Page 6, line 15: "display phases of..." or "displays a phase of..." and "This display phases of" or "This displays a phase of..."

Page 6, line 18-19: The end of the sentence is a bit strange and hard to follow.

Page 6, line 24: "A second type of root system"..., maybe it would be good to name the two types of root systems to "type 1 (horizontal) and type 2 (vertical)". I think it would be easier to follow the discussion if the first type always is horizontal and the second type always is vertical (downward) root systems. Moreover, "to speak of" doesn't read very well, maybe "horizontal root systems without any pronounced vertical root are more common at this site.

Page 6, line 25: Maybe "rising water table" is better that "higher water table".

Page 7, line 1-2: Maybe "These are interpreted as moist phases associated with raised bog expansion" is better.

Page 7, line 10-14. One aspect that could be important to regarding the tree colonization is that the colonization itself might generate positive feedback effect, which allows for further trees to establish. Evapotranspiration can generate dryer peat surface conditions, which favours further establishment of trees. This is something, which has been discussed in e.g. "Limpens, Juul, et al. 2014. How does tree density affect water loss of peatlands? A mesocosm experiment. PloS one 9.3 (experiment on tree saplings)", "Edvardsson, et al. 2015. Increased tree establishment in Lithuanian peat bogs—Insights from field and remotely sensed approaches." Science of the Total Environment 505, 113-120" (study on living peatland trees), and "Moir, A. K., et al. 2010. Dendrochronological evidence for a lower water-table on peatland around 3200-3000 BC from subfossil pine in northern Scotland. The Holocene 20.6, 931-942 (subfossil

trees)". I think that the influence of the trees themself should be mentioned in this section.

Page 7, line 13: I think "Tree colonization events (germination phases) that took place simultaneously in several different peat bogs sometimes coincide with. . ." reads better.

Page 8, line 2: TOMO_south, sometimes in italics sometimes not.

Page 8, line 7-9: Changes in the preservation conditions is also important to mention. There might, for example, have been trees growing at the mineral soil for thousands of years, but only the trees growing at the site during the expansion of the peat bog have been preserved in the moist and anaerobic conditions the water saturated zone in the bog offering have been preserved.

Page 9, line 5-7: Is there a possibility that the second, third etc. phases of tree establishment, when trees are establishing on top of root/stump layers is not as good indicators of climate/hydrology changes as the first layer? The older generations of trees will generate more stable conditions for the following generations to establish on. This could be good to expand to some extent.

Page 9, line 10: Is the brown moss is deposit directly on the mineral soil? If so, was there no lake stage in this part of the mire? Or is there a hiatus?

Page 9, line 28: Maybe "conditions unfavourable for moss growth" is better than "bog growth"

Page 10: The same information is more or less written twice in line 1-7 and line 15-20.

Page 10, line 25: What does the predating of the 8.2k event indicate? Are the trees so sensitive that they die before the actual climate change?

Page 10, line 32: I think that "The onset of the event is contemporaneous to. . ." is better than "Its beginning is. . .".

Page 11, line 18: I suggest, "The third phase of high water levels described by Magny

et al. (2004) does not. . ."

Page 11, line 25: "in" instead of "i n".

Page 12, line 1-4: Isn't the study by Dreslerova (2012) based on 14C-dated results? If so, these results might not pre-date the die-off phases, it might be within the error-bars.

FIGURES

Figure 1. I think that the figure would be improved significantly if there is a "1b" figure to the left showing a close up of the peat bog with areas with and without tree stumps, the location of the adjacent lake, the maximum extent of the lake etc.

Figure 11. I think that it would be good if germination and dying-off phases were highlighted somehow, e.g. with lines or arrows.

---

## Author Comment (AC1) · 7 Jun 2017

Response to Referee #2

Thank you very much for your time and your many valuable suggestions! I will change the manuscript according to your suggestions where possible. Please find below the response to the individual points you raised.

1. For most trees, the roots are not individually dated. I will make it more clear in the text. Root morphology has been documented for many ex situ finds at the site, giving

a good over-all impression. The reconstruction of bog expansion, which is the focus of this paper, is based in in situ finds, however. For the in situ finds, digging under the root plate has been performed for 18 tree stumps on the site, not all of which are dated unfortunately. Root morphology, described in more detail at a comparable site by Eckstein et al. 2011, served as one indicator for the main cause of tree death being water table rise. Other possible causes of cumulative tree death are discussed (and dismissed) in Eckstein et al. 2009.

2. I will get a native speaker language proofreading.

3. I will rephrase the text to make it more clear. I do believe the data to be ok, and for it to represent a climatic signal. It is a case study however, using trees grown on the site itself. It therefore represents a local signal. I therefore think it unsurprising, that the deducted hydrological variations are not throughout the exact same as those described for records located a couple of hundred kilometres away. This does not make the data less valid. Studies including several sites (e.g. Eckstein et al. 2011, Achterberg et al. 2015), or using, for example, pollen influx from a wide catchment area, give a more regional picture. I believe this study to gain value from its relatively high temporal resolution (which is possible via dendro-dates). This is further amplified by its local restriction. I highly appreciate your interest in a more detailed investigation of the individual die-off events and their environmental indication by including extensive root-related data. It should be considered for future research projects, to acquire corresponding data. I will change the text regarding your suggestion to explain better, during what conditions the bog-tree record works well, and where its limitations are.

Abstract. a) As suggested, I will include more results in the abstract. b) I will evaluate, in how far the root data can be described more in the text and abstract (compare above, 1.).

Page 1, line 11: you suggest changing 'site chronology containing gaps' to 'site chronologies'. This would regard the whole manuscript, as I speak of 'site chronology' and 'chronology segments'. I will evaluate, if the suggested change can be made without making the result more confusing in other sections.

Page 1, line 14: change will be made accordingly.

Introduction.

Page 2, line 2: I will add 'sylvestris'.

Page 2, line 3: change will be made accordingly.

Page 2, line 9: Yes, I agree. I have used the mentioned paper (Edvardsson et al. 2016) in my work after the submission of the manuscript, and will now include it in the manuscript. I will also see where I can bring in the second mentioned paper (Krapiec et al. 2016), thank you for the suggestions!

Page 2, line 13: thank you for the correction. It will be changed.

Material and Methods.

Page 2, line 20-23: I will change the sentence in close approximation to your suggestion. It actually is sand, but I can apply a more general term.

Page 2, line 29: change will be made accordingly.

Page 2, line 30: change will be made accordingly.

Page 3, line 4: change will be made accordingly.

Page 3, line 6: change will be made accordingly.

Page 3, line 6: 's.s.' is short for 'sensu stricto' meaning 'in the stricter sense'. I also made use of 's.l.' for 'sensu lato' meaning 'in the wider sense'. I will rephrase (not using those terms).

Results.

Page 3, line 25: Somewhere over a hundred. A first sampling of about 70 trees had

not delivered absolute dating. As the trees are sampled, measured and cross-dated in bundles, I could not name the exact number of trees that were necessary to acquire the first absolute dated chronology segment at the site. I can make mention of the first, unsuccessful set, though.

Page 5, line 20: change will be made accordingly.

Page 5, line 25: change will be made accordingly.

Discussion.

Page 6, line 10: change will be made accordingly.

Page 6, line 15: do you mean "the data display phases of..." would be better than "the data displays phases of..."? I will work on improved phrasing.

Page 6, line 18-19: thank you for pointing it out, I will rephrase.

Page 6, line 24: change will be made accordingly.

Page 6, line 25: change will be made accordingly.

Page 7, line 1-2: change will be made.

Page 7, line 10-14: change will be made accordingly.

Page 7, line 13: I will rephrase.

Page 8, line 2: change will be made accordingly.

Page 8, line 7-9: you are correct, the growth of trees on the mineral ground surrounding the bog is possible, and also evident (according, for example, to my own pollen data). I will include this aspect in the text.

Page 9, line 5-7: this may be true. To clarify for in situ stumps, whether they are standing on top an older tree stump however, would require a lot of digging. At the moment I don't think I can apply the theory in a good way to the present data set, but I

will give it some more thought and testing.

Page 9, line 10: yes, the brown moss peat was found directly on the sand. With the exception of two cores taken at places with relatively high elevation mineral base. I had erased the discussion about this from earlier drafts, because it was too speculative. I cannot clarify on base of our data, whether there might have been a hiatus, or if there was no, or only a brief lake stage at the site.

Page 9, line 28: change will be made accordingly.

Page 10: the text will be changed accordingly.

Page 10, line 25: that would mean, that it is uncertain, whether the die-off phase relates to the 8.2k event. I will rephrase.

Page 10, line 32: change will be made accordingly.

Page 11, line 18: change will be made accordingly.

Page 11, line 25: will be corrected.

Page 12, line 1-4: Dreslerova (2012) includes results of numerous studies which use various proxies. Dendrochronological studies are also taken into account there. I will evaluate the possibility you suggest in detail and rephrase accordingly.

Figures.

Figure 1: I agree that such a figure would be good. The required date is not available though. a) The tree stump layer is typically exposed at the terminal level of peat mining. At other places there are still peat layers of sometimes several meters thickness above the fen-bog-transition level (where extended tree layers occur typically), and modern tree growth on top of that. It would require large scale survey with high-tech support to clarify the actual extent of the tree layers below, if it is possible at all. b) I am not aware of a reconstruction of the maximum extent of lake Steinhuder Meer having been published. If will search for such information again, and if it can be found, I will gladly

include it.

Figure 11: the die-off phases are already highlighted by blue vertical bars. I had excluded the germination phases to make the figure easier to read. I will reconsider.

I also thank you for showing real interest in the research we did!

---

## Author Comment (AC2) · 7 Jun 2017

Response to Referee #1

Thank you for your time and effort! The aspects you pointed out are addressed below.

- I will revise the manuscript to make the text more clear.

- As suggested, I will get a native speaker proofreading.

- I had meant to point out that the investigated signal (water table rise) is not reflecting precipitation directly/alone, but also is affected by the variations of water-output (evapotranspiration). I will rephrase it, to make it more clear.

- I will point out the implications of the study for the understanding of past climate more clearly.

- You state that "The discussion on past climate changes seems to be quite general, and too marginal compared to the aspects of peat development.". → I will revise the section. → Even though I am not certain what exactly your statement about comparison of the discussion of climate change with aspects of peat development is aiming at (Do you suggest (a) comparison with other peat-related studies? Or (b) more extended evaluation of climatic indications in our peat stratigraphical data? Or (c) a more extended discussion of the ecological context and succession? Or (d) a more extended discussion of the climatic conditions affecting conservation?), I will pay more attention to aspects of peat development in the discussion.

- As suggested, I will include the possibility of Neolithic anthropogenic influence in the discussion. The largely contemporaneous occurrence of the tree die-off phases in the various bogs of the region (e.g. Eckstein et al. 2011, Achterberg et al. 2015) strongly supports the interpretation of a climatic trigger being most relevant, though.

---

## Author Response (AR1)

Dear CP,

- Most changes of phrasing suggested by referee #2 have been applied.
- A mentioning of the hydrological effect of tree growth on mire surface has been included, but not in section 4.3, as suggested by referee #2, but in section 4.1. (Moir et al. 2010, Limpens et al. 2014).
- Abstract has been changed (it was shortened and more results were added)
- more paleoenvironmental records were included for comparison (Edvarsson et al. 2016, Nicolussi et al. 2009, Nicolusse & Schlüchter 2012, Seppä et al. 2005), and the discussion of alignment of the climatic indications was extended. (4.7)
- As asked by referee #1 I have included a statement on the possible influence of anthropogenic clearance activity in the catchment area on the bogs hydrology. (4.7)
- As asked by referee #2 I do not speak of one "site chronology" with gaps any more, but rather of "gaps between site chronologies" (also: "gaps between site chronology segments"). I have retained the expressions "site chronology segment" and "site chronology gap" at several places in the text however. I hope this not making it more confusing.
- As asked by referee #2 I have included a statement about trees outside the bog being present but not preserved (section 4.1). (Shumilovskikh et al. 2015, Schlütz unpublished data, Behre 2008).
- The manuscript has undergone a proofreading by a native speaker

- You suggest " to show the original lake level and tree ring curves (e.g., Magny et al., 2004, Spurk et al., 2002 )". In Magny et al.2004 there is no lake level curve, but a compilation of 180 dates (of various nature) from 26 lakes. Neither is there a tree ring curve in Spurk et al. 2002, which is focused on tree deposition. I did add the curve of Schmidt et al. 2004.

Not changed:
- Referee #2 had suggested: „Page 6, line 25: Maybe "rising water table" is better that "higher water table".", which I had agreed to comply to in my answer. However, I am of the opinion, that the water table being higher (than in other time-intervals) is not the same as the water table rising (which implies a change within the time-interval discussed). I therefore have decided not to change the text in this respect.

Added refernces:

- Edvardsson, J., Stoffel, M., Corona, C., Bragazza, L., Leuschner, H.H., Charman, D. J., Helama, S.: Subfossil peatland trees as proxies for Holocene palaeohydrology and palaeoclimate. Earth- Science Reviews 163, 118-140, 2016.

- Holmgren, M., Lin, C.-Y., Murillo, J., Nieuwenhuis, A., Penninkhof, J., Sanders, N., van Bart, T., van Veen, H., Vasander, H., Vollebregt, M., Limpens, J.: Positive shrub-tree interactions facilitate woody encroachment in boreal peatlands. Journal of Ecology, 103, 58-66, 2015.doi: 10.1111/1365-2745.12331

- Krąpiec, M., Margielewski, W., Korzenń, K., Szychowska-Krąpiec, E., Łajczak, A.: Late Holocene palaeoclimate variability: The significance of bog pine dendrochronology related to peat stratigraphy. The Puścizna Wielka raised bog case study (Orawa – Nowy Targ Basin, Polish Inner Carpathians). Quaternary Science Reviews, 148, 192-208, 2016.

- Limpens, J., Holmgren, M., Jacobs, C. , Van de Zee, S., Karofeld, E., Berendse, F.: How does tree density affect water loss of peatlands? A mesocosm Experiment. PLOS One 9 (3), 1-11, 2014.

- Nicolussi, K. & Schlüchter, C.: The 8.2 ka event – Calendar-dated glacier response in the Alps. Geology 40 (9), 819-822, 2012.doi:10.1130/G32406.1

- Nicolussi, K., Kaufmann, M., Melvin, T. M., van der Plicht, J., Schießling, P., Thurner, A.: A 9111 year lon conifer tree-ring chronology for the European Alps: a base for environmental and climatic investigations. The Holocene 19(6), 909-920, 2009.

- Seppä, H., Hammarlund, D., Antonsson, K.: Low-frequency changes in temperature and effective humidity during the Holocene in south-central Sweden: implications for atmospheric and oceanic forcings of climate. Climate Dynamics 25, 285-297, 2005. DOI 10.1007/s00382-005-0024-5

- Shumilovskikh, L., Schlütz, F., Achterberg, I., Kvitkina, A., Bauerochse, A., Leuschner, H.H.: Pollen as a nutrient source in Holocene ombrotrophic bogs. Review of Palaeobotany and Palynology 221, 171- 178, 2015.

I thank you very much for your work with our manuscript.
I hope you find the changes agreeable.

With best regards,
Inke Achterberg

[revised manuscript text omitted]

---

## Author Response (AR2)

Dear CP,

I sincerely thank the editor for his efforts to improve this manuscript.

I have revised the Figure 11, which is now split into two figures (Figure 11 A and B).

Figure 11 A shows the River Main oak deposition curve (which I was asked to add), the lake level plot by Magny et al. 2004 (which I was asked to add), and the replication of the *TOMO_south* pine chronologies and the total replication of the northwest German pine chronology (which *TOMO_south* is part of).

To section 4.7 an paragraph describing the comparisons of Figure 11 A was added.

Figure 11 B mostly shows the same comparisons of *TOMO_south* phases with other studies as in the previous version, but the replication curves were removed from the figure and some graphic changes were made, which hopefully improve readability.

I hope you consider the applied changes an improvement.

With the best regards,

Inke Achterberg

[revised manuscript text omitted]

---

## Author Response (AR3)

Dear Editor,

thank you for your efforts regarding this manuscript.

However, the editorial board informed me, that there would be language correction included in the final processing of all accepted papers. I was told, I therefore would not have to commission a second native speaker proofreading on my own expense. I therefore did not commission a new native speaker proofreading, but only looked for mistakes myself. Some spelling mistakes were corrected. I hope this, together with the proofreading commissioned by the journal, will be satisfactory.

With kind regards,

Inke Achterberg

[revised manuscript text omitted]